# Reliability Assessment of Reinforced Concrete Beams under Elevated Temperatures: A Probabilistic Approach Using Finite Element and Physical Models

**János Szép** [1], **Muayad Habashneh** [1], **János Lógó** [2,3] and **Majid Movahedi Rad** [1,*]

1   Department of Structural and Geotechnical Engineering, Széchenyi István University, H-9026 Gyor, Hungary
2   Department of Structural Mechanics, Budapest University of Technology and Economics,
    H-1111 Budapest, Hungary
3   Department of Highway and Railway Engineering, Budapest University of Technology and Economics,
    H-1111 Budapest, Hungary
*   Correspondence: majidmr@sze.hu

**Abstract:** A novel computational model is proposed in this paper considering reliability analysis in the modelling of reinforced concrete beams at elevated temperatures, by assuming that concrete and steel materials have random mechanical properties in which those properties are treated as random variables following a normal distribution. Accordingly, the reliability index is successfully used as a constraint to restrain the modelling process. A concrete damage plasticity constitutive model is utilized in this paper for the numerical models, and it was validated according to those data which were gained from laboratory tests. Detailed comparisons between the models according to different temperatures in the case of deterministic designs are proposed to show the effect of increasing the temperature on the models. Other comparisons are proposed in the case of probabilistic designs to distinguish the difference between deterministic and reliability-based designs. The procedure of introducing the reliability analysis of the nonlinear problems is proposed by a nonlinear code considering different reliability index values for each temperature case. The results of the proposed work have efficiently shown how considering uncertainties and their related parameters plays a critical role in the modelling of reinforced concrete beams at elevated temperatures, especially in the case of high temperatures.

**Keywords:** reinforced concrete beams; reliability analysis; elevated temperature; compressive strength; mechanical properties

## 1. Introduction

Concrete can be considered as one of the mostly used materials in construction projects due to some important aspects, such as its strength, durability, and fire resistance. Several studies have been conducted to investigate the behavior of reinforced concrete beams under different loading and environmental conditions. A new shear reinforcement configuration was proposed by Demir et al. [1] by adapting a nonlinear finite element study to investigate the improvement of shear capacity in reinforced concrete beams. Salih and Zhou [2] studied the behavior of concrete beams reinforced with fiber polymer bars by conducting a series of finite element models, while Kytinou et al. [3] analyzed the steel fiber's effects on the flexural performance of steel fiber-reinforced concrete. Chalioris et al. [4] developed a new structural health monitoring system by demonstrating its use and efficiency. This method could evaluate structural damage caused by concrete cracking and steel yielding under monotonic and cyclic loading. In addition, by using ferrocement composite as transverse reinforcement, Megarsa and Kenea [5] presented the results of a comprehensive analytical examination of the beams' shear performance. Kytinou et al. [6] performed numerical analysis using ABAQUS 2018 finite element software for predicting the cyclic

lateral response of reinforced concrete (RC) beam–column connections using composite carbon fiber-reinforced polymer (CFRP) bars as a longitudinal reinforcement in the beam.

However, in the cases of high temperatures, concrete's mechanical properties are changed. Thus, in the case of elevated temperatures, failure might occur due to the presence of cracks which are parallel to the heat surface [7–12]. Consequently, concrete properties after fire exposure are still significant for estimating the load-carrying capacity [12,13]. Furthermore, it has been concluded that under high temperatures, the changes in chemical and physical properties of the concrete are not only dependent on the matrix composition, but also on the water/cement ratio, age of concrete, and the aggregate's type [14–16].

In fact, the idea of investigating the mechanical properties of concrete at elevated temperatures has attracted many researchers recently, which was resulted in different experimental tests [17–20].

Carbon fiber-reinforced polymer (CFRP) bars were used in the reinforcement of a concrete beam, and the behaviour of the beam was numerically examined while being subjected to high temperatures in the study of Ilango and Mahato [21]. Kakae et al. [22] investigated the obtained physical properties of concrete, such as Young's modulus, compressive strength and strain, and thermal properties, when it was exposed to high temperatures. Xiao and König [23] presented a general overview of the mechanical behavior of concrete when it is exposed to high temperatures by making a comparative analysis of prior research. By utilizing thermo-chemical reactions of the concrete, Cioni et al. [24] presented an assessment of fire damage of reinforced concrete elements. In addition, by replacing cement with finely ground pumice (FGP) of the concrete mix with different proportions of weight, the properties of concrete were investigated at elevated temperatures by Demirel and Keleştemur [25]. Savva et al. [26] also showed in their study how high temperatures affect the mechanical properties of pozzolanic concrete, in which the findings of the study concluded that the concrete's residual properties crucially depend on the type of aggregates and binder. By considering different heating loads, the mechanical properties of residual fracture, such as stress intensity factor and fracture energy, were analyzed by Hlavička et al. [27]. Song et al. [28] determined that longitudinal reinforcement and proper stirrup reinforcement designs may enhance the fire resistance performance of simply supported reinforced concrete beams. Fire testing followed by bending tests on reinforced concrete specimens were conducted by Cai et al. [29], where the proposed work showed a sufficient agreement between the theoretical calculation and FEA results. Furthermore, Agrawal and Kodur [30] proposed valuable information about the residual capacity of high-strength concrete beams after exposure to fire, indicating that such beams can recover a significant portion of their flexural capacity provided they survive the fire exposure. The effects of fire exposure on the residual shear and flexural behavior of reinforced concrete beams were analyzed by Yuye et al. [31], who proposed a practical calculation method for assessing shear performance after fire.

Li and Purkiss [32] investigated the mechanical properties of concrete at elevated temperatures in their study, and one of their findings was that stress–strain diagrams in EN 1992-1-2 [33] are risky in the existence of high axial loads due to high peak strains. Furthermore, Kim et al. [34] tested the impact of high temperatures on concrete's strength, and they contrasted their findings with those of ACI [35] model values.

Additionally, there are several research works which were conducted to study the effect of temperatures on the properties of reinforcing steels. For instance, Felicetti et al. [36] investigated how steel bars react after being exposed to high temperatures. Furthermore, Dotreppe [37] showed in his study how quenched and self-tempered steels' mechanical characteristics may be impacted by temperature.

By considering the aim of structural engineering of proposing structural models that meet the safety and serviceability conditions, uncertainties that might be related to the material properties and the applied loading conditions should be considered in the design process [38–41]. Thus, the reliability-based algorithm was introduced into deterministic designs of concrete structures [42,43]. Rakoczy and Nowak [44] presented

a reliability analysis for prestressed concrete where sensitivity functions were developed to consider the effect of reliability indices on the concrete girders. Reinforced concrete beams were considered for the reliability analysis in the study of Słowik et al. [45], where the experimental findings were used to determine the safety margins of the designed shear resistance. Additionally, Olmati et al. [46] introduced a framework of probabilistic design for flat slab punching due to accidental loads, such as a slab falling from above, column removal, or a blast load. By taking into account uncertainties when analyzing reinforced concrete beams, Schlune et al. [47] used the reliability level in which a safety format was proposed based on the nonlinear analysis and a resistance safety factor. Eamon and Jensen [48] explained in detail a way to figure out how reliable RC beams are under a fire load by assuming concrete compressive strength, steel yielding strength, and other important parameters as random variables. Li et al. [49] provided valuable insights into the reliability of HSC beams under elevated temperatures that could inform the design and assessment of these structures in fire conditions.

Many researchers have investigated the physical and mechanical properties of concrete at high temperatures, including Young's modulus, compressive strength, and strain. However, the variability of material properties has created uncertainties in the results of previous research.

To address this issue, this paper aims to examine the effect of considering reliability design in the numerical analysis of reinforced concrete beams at elevated temperatures. The proposed work seeks to provide insights into the behavior of concrete and its reliability under high temperatures, informing the design and assessment of concrete structures in fire conditions. Taking into consideration that the proposed work in this paper considers EN 1992-1-2 [33] to show the proficiency of the proposed method. In addition, to achieve the desired aim, a written nonlinear programming code is developed to perform the reliability analysis by considering that the introduced reliability index ($\beta$) plays as a limit when the concrete properties are considered as random variables which follow normal distribution. Moreover, the Monte Carlo sampling method is considered in order to determine reliability indices based on the statistics of the concrete properties.

The rest of the paper is constructed as follows. Section 2 provides an overview of the reliability analysis. The considered constitutive concrete model is presented in Section 3. Section 4 demonstrates the experimental program of the considered reinforced concrete beam, while the numerical modelling including the model validation, considering the effect of elevated temperatures and introducing the reliability design, are presented in Section 5. Lastly, Section 6 includes the conclusions and remarks of the proposed work.

## 2. Reliability Analysis

By recalling the basic concept of reliability analysis and assuming that $X_R$ stands for $X_S$ non-negative bound, the failure is defined by $X_R \leq X_S$, taking into consideration that $X_S$ and $X_R$ are independent random variables in which their probability density functions are $f_R(X_S)$ and $f_R(X_R)$, respectively. Therefore, for the estimation failure probability, the following equation is used [50]:

$$P_f = P[X_R \leq X_S] = \iint_{X_R \leq X_S} f_R(X_R) f_S(X_S) dX_R dX_S \tag{1}$$

$$= \int_0^\infty F_R(X_S) f_S(X_S) dX_S; \tag{2}$$

$$= \int_0^\infty [1 - F_S(X_R)] f_R(X_R) dX_R = 1 - \int_0^\infty F_S(X_R) f_R(X_R) dX_R; \tag{3}$$

where $F_R(X_R) = \int_0^{X_R} f_R(t)dt$ and $F_S(X_S) = \int_0^{X_S} f_S(t)dt$ denote the cumulative distribution functions (CDFs) of $X_R$ and $X_S$, respectively. For most distributions of $X_R$ and $X_S$, the above integrals will have to be evaluated numerically.

Equation (1) can be alternatively written in the case of considering limit state function, as follows:

$$g(X_R, X_S) = X_R - X_S \tag{4}$$

where the failure domain $D_f$ is described by $g \leq 0$. Hence, $P_f$ can be determined by:

$$P_f = F_g(0) \tag{5}$$

In fact, $P_f$ can be expressed as follows:

$$P_f = \int_{g(X_R, X_S) \leq 0} f(X)dX = \int_{D_f} f(X)dX \tag{6}$$

The Monte Carlo sampling technique is considered in this study for estimating $P_f$. The very basic concept of this technique is about generating the expression $x$ of the random vector $X$ depending on probability density function $f_X(x)$. By determining the ratio of points inside the $D_f$ to the total generated points, $P_f$ can be calculated using the Monte Carlo technique. This assumption is formulated by writing an equation considering indicator function of $D_f$, as follows:

$$\chi_{D_f}(x) = \begin{Bmatrix} 1 \ if \ x \in D_f \\ 0 \ if \ x \notin D_f \end{Bmatrix} \tag{7}$$

Accordingly, $P_f$ can be rewritten as follows:

$$P_f = \int_{-\infty}^{+\infty} \cdots \int_{-\infty}^{+\infty} \chi_{D_f}(x) f_X(x)dx \tag{8}$$

The distribution of random variable $\chi_{D_f}(X)$ points are as follows:

$$\mathbb{P}\left[\chi_{D_f}(X) = 1\right] = P_f \tag{9}$$

$$\mathbb{P}\left[\chi_{D_f}(X) = 0\right] = 1 - P_f \tag{10}$$

where $P_f = \mathbb{P}\left[X \in D_f\right]$. Bearing in mind that $\chi_{D_f}(X)$ is random variable following normal distribution which has mean value and variance:

$$\mathbb{E}\left[\chi_{D_f}(X)\right] = 1 \cdot P_f + 0 \cdot \left(1 - P_f\right) = P_f \tag{11}$$

$$\mathbb{V}ar\left[\chi_{D_f}(X)\right] = \mathbb{E}\left[\chi_{D_f}^2(X)\right] - \left(\mathbb{E}\left[\chi_{D_f}(X)\right]\right)^2 = P_f - P_f^2 = P_f\left(1 - P_f\right) \tag{12}$$

To calculate $P_f$, the estimator of the mean value is utilized as follows:

$$\hat{\mathbb{E}}\left[\chi_{D_f}(X)\right] = \frac{1}{Z}\sum_{z=1}^{Z} \chi_{D_f}\left(X^{(z)}\right) = \hat{P}_f \tag{13}$$

where $X^{(z)}$ indicates the independent random vectors, and $z = 1, \ldots, Z$ are accompanied with probability density functions.

In order to consider uncertainties in our work, material properties of concrete and steel are assumed as random variables following a Gaussian distribution where the estimator of the mean value ($\mathbb{E}$) and variance ($\mathbb{V}ar$) are determined as follows:

$$\mathbb{E}\left[\hat{P}_f\right] = \frac{1}{Z}\sum_{z=1}^{Z} \mathbb{E}\left[\chi_{D_f}\left(X^{(z)}\right)\right] = \frac{1}{Z}ZP_f = P_f \tag{14}$$

$$\mathbb{V}ar\left[\hat{P}_f\right] = \frac{1}{Z^2}\sum_{z=1}^{Z} \mathbb{V}ar\left[\chi_{D_f}\left(X^{(z)}\right)\right] = \frac{1}{Z^2}ZP_f\left(1 - P_f\right) = \frac{1}{Z}P_f\left(1 - P_f\right) \tag{15}$$

Because of some difficulties in the process of computing $P_f$ accurately, the first-order reliability methods were utilized, where the reliability index ($\beta$) is used. The benefits of utilizing ($\beta$) are that as there are many applications of reliability analysis in engineering designs, the desired ($\beta$) controls more regular engineering practices and, thus, engineering standards, especially structural ones, offer an extensive set of target values (e.g., EN1990 [51]).

By utilizing ($\beta$), the reliability limit can be constructed as follows:

$$\beta_{target} - \beta_{calc} \leq 0 \qquad (16)$$

Lastly, the following expressions are used to determine $\beta_{target}$ and $\beta_{calc}$:

$$\beta_{target} = -\Phi^{-1}\left(P_{f,target}\right) \qquad (17)$$

$$\beta_{calc} = -\Phi^{-1}\left(P_{f,calc}\right) \qquad (18)$$

## 3. The Adopted Constitutive Model

In the accessible scientific papers, the explanations of this model can be deeply tracked. However, a short and brief description is introduced here. By adopting the Prandtl–Reuss concept regarding the elastoplastic deformation, the overall strain tensor $\varepsilon_{ij}$ consists of an elastic part ($\varepsilon_{ij}^{el}$) and a plastic one ($\varepsilon_{ij}^{pl}$) as explained in Equation (19):

$$\varepsilon_{ij} = \varepsilon_{ij}^{el} + \varepsilon_{ij}^{pl} \qquad (19)$$

Furthermore, the scalar damage elasticity formula governs the internal stress–strain relationships as follows:

$$\hat{\sigma}_{ij} = D_{ijkl}^{el} \times \left(\varepsilon_{ij} - \varepsilon_{ij}^{pl}\right) \qquad (20)$$

where $D_{ijkl}^{el}$, indicating the degraded elastic stiffness, is written as follows:

$$D_{ijkl}^{el} = (1-d)D_0^{el} \qquad (21)$$

where $D_0^{el}$ represents the initial (elastic) stiffness of the material, and $d$ indicates the stiffness degradation.

It is important to take into consideration that $d$ might vary from (0) in the case of undamaged material to (1) in the case of fully damaged material. The stiffness reduction here is isotropic, taking into consideration that a single variable of degradation ($d$) is used to describe it. According to the concept of damage mechanics of continuum structures, the effective internal force $\left(\overline{\sigma}_{ij}\right)$ is described as follows:

$$\overline{\sigma}_{ij} = D_0^{el} \times \left(\varepsilon_{ij} - \varepsilon_{ij}^{pl}\right) \qquad (22)$$

The relationship between the internal force $\hat{\sigma}_{ij}$ and the effective internal force $\overline{\sigma}_{ij}$ by adopting scalar reduction relation can be constructed as follows:

$$\hat{\sigma}_{ij} = (1-d) \cdot \overline{\sigma}_{ij} \qquad (23)$$

In the case of $d = 0$, $\hat{\sigma}_{ij} = \overline{\sigma}_{ij}$. Nonetheless, the effective internal force turns out to be more illustrative than internal force due to the resistance of the external loads by the area of the effective internal force. By considering the nominal stress and the reduced elastic tensor which is shown in Equation (22), Equation (20) can be rewritten as follows:

$$\hat{\sigma}_{ij} = (1-d)D_0^{el} * \left(\varepsilon_{ij} - \varepsilon_{ij}^{pl}\right) \qquad (24)$$

The constitutive model of damaged plasticity is illustrated by the internal force–strain relationship:

$$\hat{\sigma}_{ij} = (1-d) \cdot \overline{\sigma}_{ij} \rightarrow \hat{\sigma}_{ij} = (1-d_t)\overline{\sigma}_{t_{ij}} + (1-d_c)\overline{\sigma}_{C_{ij}} \tag{25}$$

where $d_c$ and $d_t$ represent the variables of compression and tension damage, respectively, taking into consideration that these variables are varying from 0 to 1 for undamaged and fully damaged cases, respectively. Furthermore, $\overline{\sigma}_t$ is the effective tension internal force and $\overline{\sigma}_c$ represents the effective compression internal force. In general, the damage model of concrete considers the compressive crushing and tensile cracking failures. Furthermore, it is assumed that plasticity damage affects the uniaxial compressive and tensile response of concrete, as can be seen in Figure 1; this can be calculated as follows:

$$\sigma_t = (1-d_t)E_0\left(\varepsilon_t - \varepsilon_t^{pl,h}\right) \tag{26}$$

$$\sigma_c = (1-d_c)E_0\left(\varepsilon_c - \varepsilon_c^{pl,h}\right) \tag{27}$$

where $E_0$ represents the initial Young's modulus, $\varepsilon_t^{pl,h}$ is the equivalent tension plastic strain, and $\varepsilon_c^{pl,h}$ is the equivalent compression plastic strains. Therefore, the effective uniaxial compressive $\overline{\sigma}_c$ and tensile $\overline{\sigma}_t$ stresses are computed as follows:

$$\overline{\sigma}_t = \frac{\sigma_t}{(1-d_t)} = E_0\left(\varepsilon_t - \varepsilon_t^{pl,h}\right) \tag{28}$$

$$\overline{\sigma}_c = \frac{\sigma_C}{(1-d_c)} = E_0\left(\varepsilon_c - \varepsilon_c^{pl,h}\right) \tag{29}$$

where tensile strain $\varepsilon_t = \varepsilon_t^{pl,h} + \varepsilon_t^{el}$, and compressive strain $\varepsilon_c = \varepsilon_c^{pl,h} + \varepsilon_c^{el}$. Thus, it can be said that $\varepsilon_t^{el}$ represents the equivalent elastic strains for tension and $\varepsilon_c^{el}$ represents the equivalent elastic strains for compression.

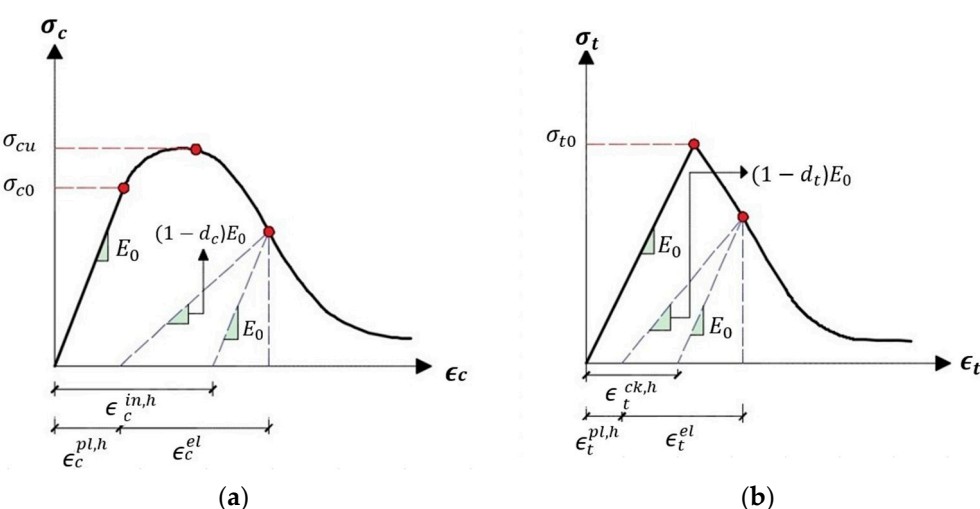

**Figure 1.** Uniaxial loading condition. (**a**) Compression; (**b**) tension.

To sum up, the concrete damage plasticity (CDP) constitutive model is adopted to show the tension and compression behaviors of concrete within the validated model.

## 4. Experimental Tests

Experimental tests of three simply supported reinforced concrete beams are utilized in this section; these tests were held in the laboratory of Széchenyi István University. Furthermore, four samples were equipped for the tests on the properties of concrete. The

standard cube test, which involves compressing a cubic sample of concrete under controlled conditions, was utilized for determining the compressive behavior of the concrete in this study, as it is a widely accepted method for measuring the compressive strength of concrete. The split cylinder test was adopted for concrete tensile behavior, taking into consideration that it is a reliable and widely accepted method for determining the tensile strength of concrete and is used in many design codes and standards for reinforced concrete structures. It was appropriate as the samples were stored at standard conditions of temperature = 20 °C.

After 28 days of curing, the samples were tested, and the compression and tension properties were assessed and shown in Figure 2, taking into consideration that Young's modulus $E_0 = 34,951$ N/mm$^2$ and Poisson's ratio $v = 0.2$. The used reinforcing steel in this study was hot-rolled B500 reinforcing steel. Taking into account that a tensile test was performed, the obtained mechanical properties of the steel reinforcement are represented in Table 1. The geometry of the beams was length = 2500 mm, with a cross-sectional area of (150 mm × 300 mm). The layout of the experiments, including the geometry, the boundary conditions, and the applied load for the considered reinforced concrete beams, is shown in Figure 3, considering that the beams were tested by applying one concentrated monotonic loading up to failure.

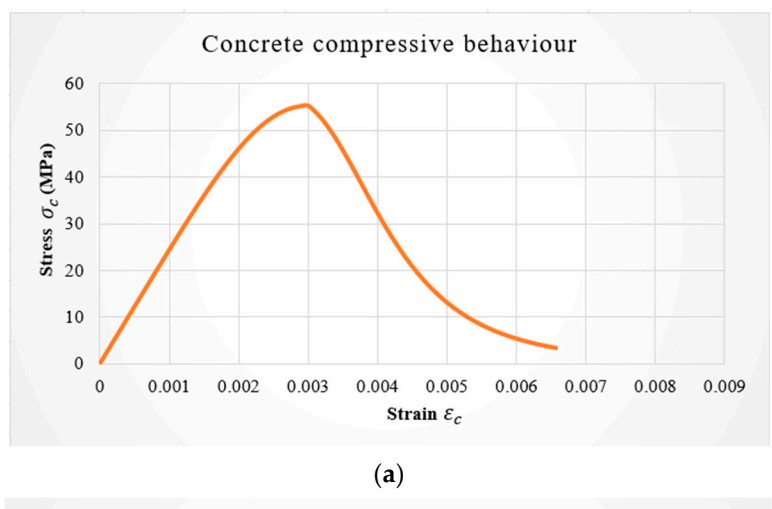

(**a**)

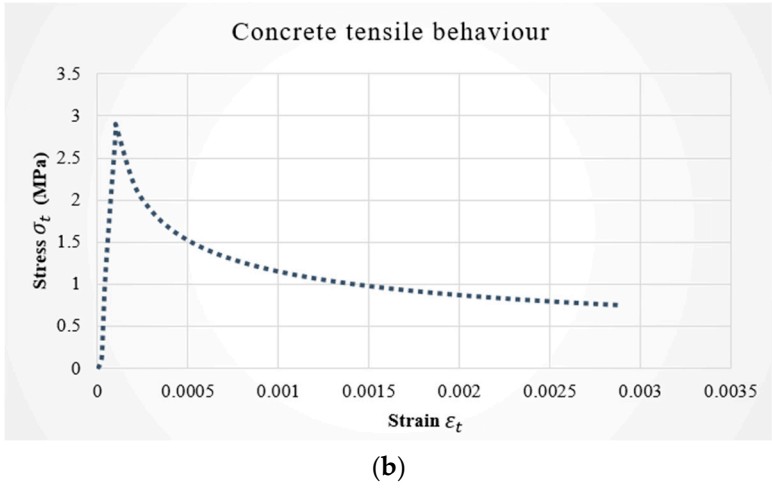

(**b**)

**Figure 2.** The obtained concrete properties. (**a**) Compression; (**b**) tension.

**Table 1.** Properties of the considered steel bars.

| Diameter (mm) | Yield Strength (MPa) | Ultimate Tensile Strength (MPa) | Young's Modulus (MPa) |
|---|---|---|---|
| φ = 12 mm | 540 | 662 | 200,000 |
| φ = 8 mm | 583 | 687 | 200,000 |

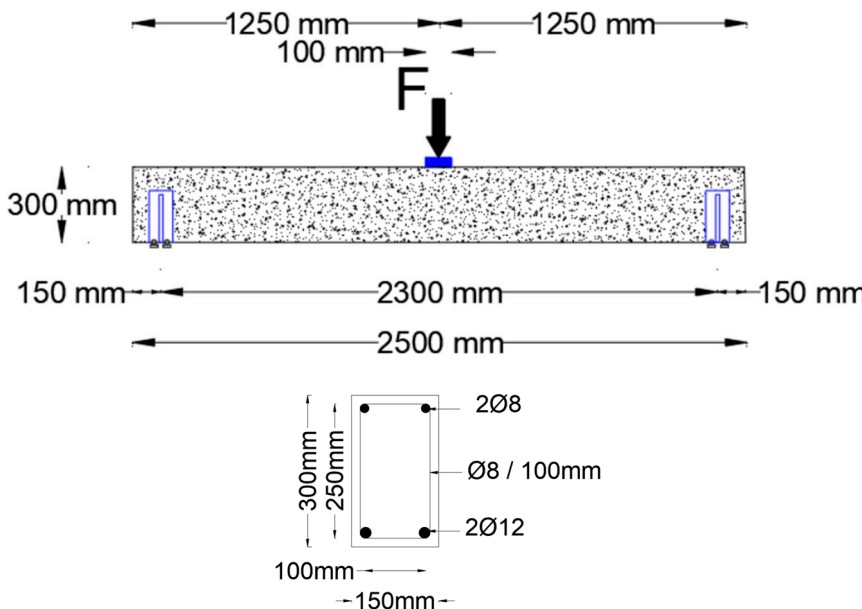

**Figure 3.** Geometry, loading condition, and cross-section of the beam.

## 5. Numerical Modelling

### 5.1. Finite Element Model

In this section, validation of the model using 3D finite element analysis (FEA) based on the experimental tests is proposed. A finite element (FE) model of the reinforced concrete (RC) beam is made for this study using the commercially available software ABAQUS® [52]. ABAQUS is a finite element analysis (FEA) software package widely used for simulating the behavior of engineering structures and materials under various loading conditions. It is capable of handling complex geometric and material models, and provides a range of analysis capabilities including structural, thermal, acoustic, and electromagnetic analysis.

ABAQUS software has the capability to predict the failure of concrete by implementing a concrete damage plasticity model for material properties. This model considers the process of damage that occurs through micro-cracking, which initially starts on a specific beam section and gradually widens before coalescing to ultimately cause failure. The plasticity behavior can be described by various phenomena, such as strain softening, gradual deterioration, and volumetric expansion, which result in a decrease in both the strength and stiffness of concrete. The degradation of stiffness is typically indicative of the damage sustained. Failure is assumed to result from two primary factors: tensile cracking and compression crushing of the concrete material.

A parametric study was carried out to figure out the right mesh size for the finite element model. As a result, the best mesh size was found to be 25 mm, in which the element mesh sizes were carefully chosen so that the result could be reached with proper computational time. The reinforced concrete beam was modelled by using eight-node first-order hexahedral (C3D8) elements for concrete, and truss elements for steel, as is shown in Figure 4. Furthermore, the geometry, boundaries, and loading conditions of the beams are represented in Figure 5. Here, a single concentrated loading was applied at the middle of the top flange and approximately 7200 elements were used to generate

the finite element mesh of the concrete, while 984 elements were used for the mesh of steel reinforcement. Furthermore, a constraint of an embedded region was applied for the purpose of simulating the bond between the reinforcements with the concrete. The nonlinear behavior and the adoption of the damage plasticity model were carried out by utilizing finite element analysis for the beams.

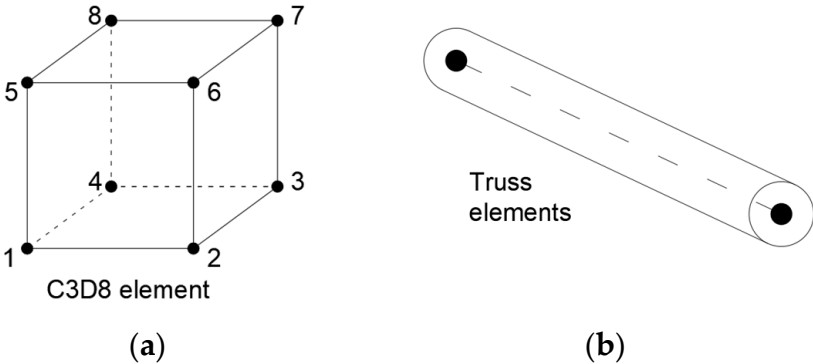

(**a**)     (**b**)

**Figure 4.** Considered elements for modelling. (**a**) Concrete modelling; (**b**) Steel modelling.

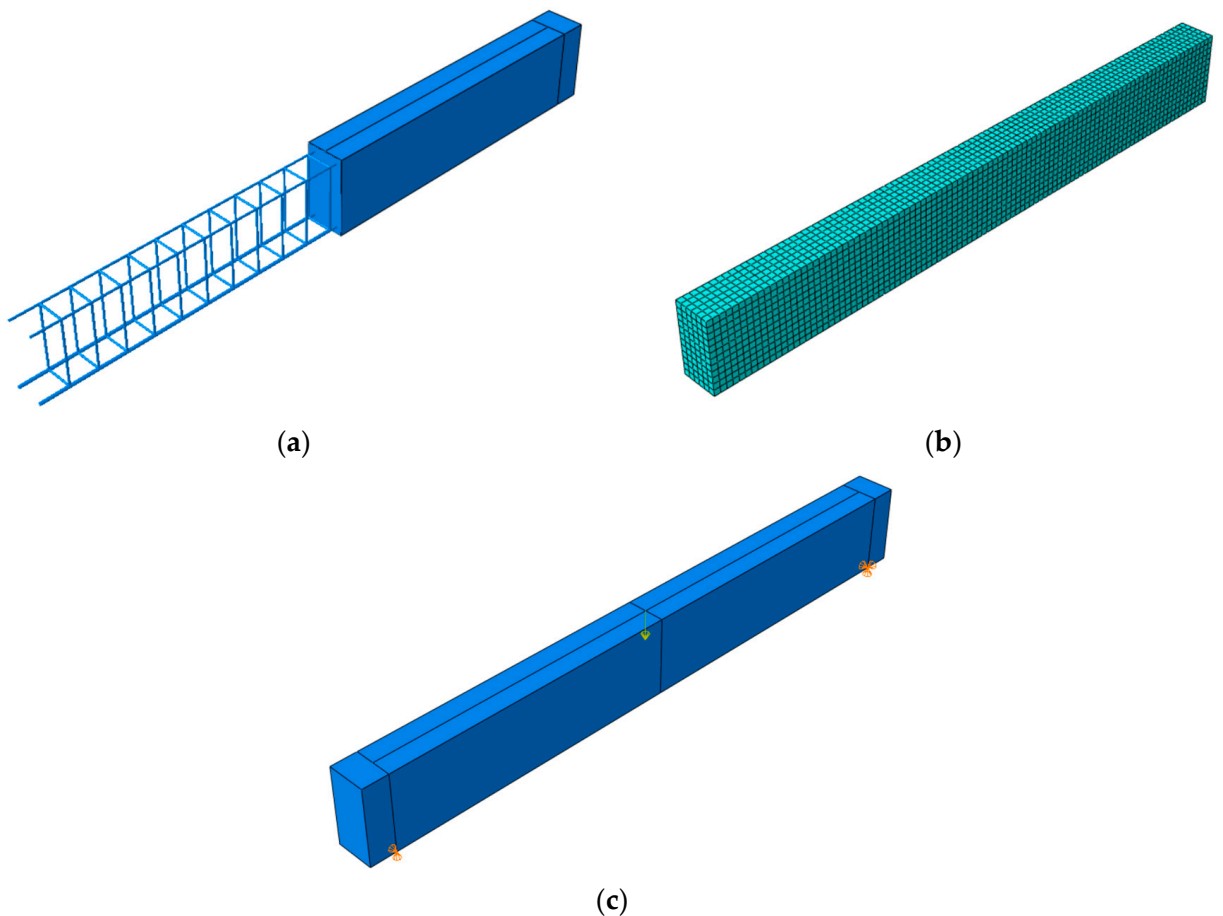

(**a**)     (**b**)

(**c**)

**Figure 5.** Numerical reinforced concrete model. (**a**) Model assembly; (**b**) finite element mesh of the model; (**c**) loading and support conditions of the model.

## 5.2. Model Validation

According to the mechanical properties of the tested specimens (Figure 2), the (CDP) data, which contains the compressive crushing, as well as the tensile cracking, were recorded, then these data were imported to the FEA 2018 software to obtain CDP pa-

rameters that reveal the damage behavior of concrete. The CDP parameters which were assumed in this study are illustrated in Table 2, considering that the CDP parameters were kept constant during the reliability-based analysis process later. Furthermore, it should be noted that in this work, a variety of dilation angle values were investigated. Only the value that best predicts the deflection response of the experimental tests is presented here.

**Table 2.** Concrete damage plasticity data.

| Dilation Angle | Eccentricity | $f_{b0}/f_{c0}$ | K |
|---|---|---|---|
| 30 | 0.1 | 1.16 | 0.667 |
| **Concrete Tensile Behavior** | | **Concrete Tension Damage** | |
| Yield stress (MPa) | Cracking Strain | Damage Parameter T | Cracking Strain |
| 2.9 | 0 | 0 | 0 |
| 0.754896741 | 0.0029 | 0.739959615 | 0.002878401 |

The numerical results of the models were verified according to the experimental results. It is worth mentioning that since the load is applied at the middle of the top flange of the beam, the damage pattern is within the area around the point of applied load which can be obviously seen from Figure 6 where the experimental model and numerical model have almost the same concrete damage patterns (tension damage), where the intensity of the damaged zones ranges from the blue color that represents the undamaged zones (*dt* = 0), to the red color which represents the fully damaged zones (*dt* = 1).

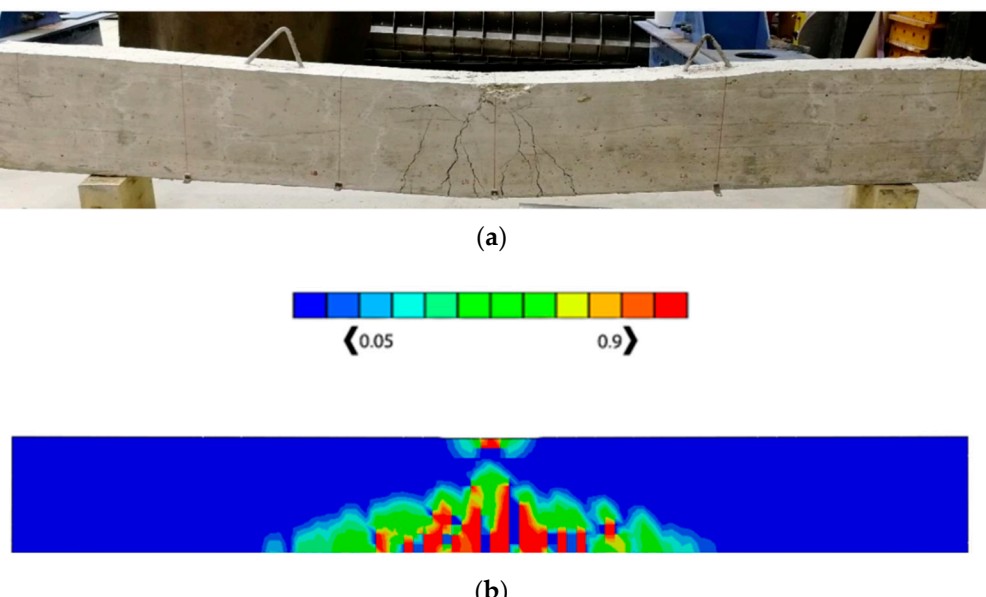

**Figure 6.** Failure mechanism of the beam. (**a**) Experimental model; (**b**) numerical model.

Moreover, load–deflection diagrams of the numerical and experimental tests are shown in Figure 7, where the measured ultimate load capacity was about 74 kN in the case of the numerical model, while it was approximately 72 kN for the average experimental results.

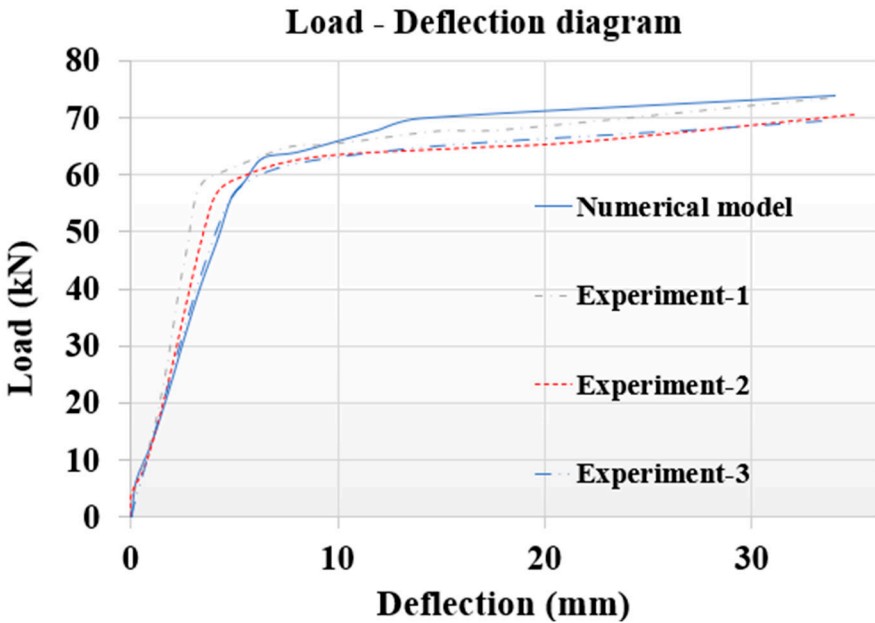

**Figure 7.** Load–deflection diagrams of numerical and experimental tests.

*5.3. Reinforced Concrete at Elevated Temperatures*

In this part, the effect of temperature is considered for the reinforced concrete beam according to the stress–strain curves which was mentioned in the study of Schneider [53] for concrete material and according to the work of Meda et al. [54] for steel. It is worth mentioning that there are many available papers and monographs which can be referred to in order to find detailed descriptions and it is an open question between researchers, but as was mentioned earlier, this study aims to validate the proposed method.

The validated reinforced concrete beam model is considered in this part, where it is modelled by the commercially FEA software ABAQUS [52] according to different temperature values which are 20 °C, 150 °C, 350 °C, 450 °C, and 750 °C, and the corresponding mechanical properties of concrete are shown in Table 3. The used bars were assumed as hot-rolled reinforcing steel, taking into consideration that the stress–strain diagrams of the steel at elevated temperatures which are represented in Figure 8 were utilized according to Eurocode [55]. The proposed material properties of concrete and steel are used in the numerical analysis for the different considered temperatures. Thus, the results of the corresponding ultimate values of load and displacements are given in Table 4, in which the CDP constitutive model is adopted. Table 4 also represents the results of stress and tension damage intensities in case of deterministic designs, taking into consideration that the value of the applied load is 36 kN (the ultimate load value in the case of temperature = 750 °C). It can be noted from Table 4 that as temperature increases, the corresponding values of ultimate load and displacements decrease. For instance, the displacement value is decreased by 75% from 33 mm in the case of 20 °C to 8.18 mm in the case of 750 °C. Additionally, the ultimate load value is decreased by 51.35% from 74 kN in the case of 20 °C to 36 kN in the case of 750 °C. Furthermore, it can obviously be shown that the yielded stress zones which are represented by the red color within the steel reinforcement and the resultant tension damaged area of concrete increase as the temperature values increase, taking into consideration that the blue color represents the undamaged areas, and the red color reflects the damaged areas. Furthermore, it can be noticed that when the temperature increases, the damage pattern and stress intensity distributions are extended away from the middle area of the model.

**Table 3.** Mechanical properties of concrete.

| Temperature (°C) | Compressive Strength—$f'c$ (MPa) | Young's Modulus (GPa) |
|---|---|---|
| 20 | 55.30 | 34.95 |
| 150 | 48.60 | 32.70 |
| 350 | 45.30 | 31.60 |
| 450 | 42.50 | 30.60 |
| 750 | 13.82 | 17.40 |

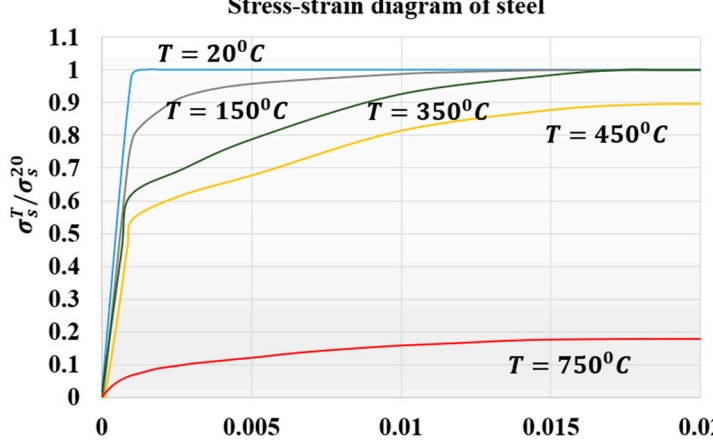

**Figure 8.** Stress–strain diagrams of steel according to Eurocode [55].

By considering the same value of the applied load ($F = 36$ kN), Table 4 represents the obtained results of the percentage of the tensile damaged elements within the model for each temperature case. It should be noted that the percentage increases as the temperature increases. For instance, the percentage value is increased from 3.16% in the case of temperature = 20 °C to 41% in the case of temperature = 750 °C.

### 5.4. Introducing Reliability Analysis

The properties of the concrete and steel materials are selected as the parameters affecting the performance of the reinforced concrete beam at increased temperatures. Due to the unpredictability inherent in the microstructure of the material, it is possible that the stress–strain curves for the same material strength cannot be replicated exactly, resulting in a considerable variation in the real capacity of the structure. To obtain a more accurate response for the compressive strength and elasticity modulus of concrete, some researchers have used the normal or lognormal distribution as the best match.

As was mentioned previously, a nonlinear code was written to perform the reliability analysis by considering that the introduced reliability index governs the process as well as playing a role as a limit by assuming the concrete and steel properties which were given in Table 3 and Figure 8 (Section 5.3) as random variables following a normal distribution. In order to determine ($\beta$), the Monte Carlo method is used by considering that the total samples number ($Z = 3 \times 10^8$). The considered concrete material's probabilistic parameters are shown in Table 5. Furthermore, the deterministic steel material's properties, which were mentioned earlier in Figure 8, are considered as mean values with 5% standard deviation for the probabilistic case. It is worth mentioning that the corresponding CDP parameters are changed accordingly.

**Table 4.** Maximum load capacities, displacements, and stress and tensile damage intensity of the beam.

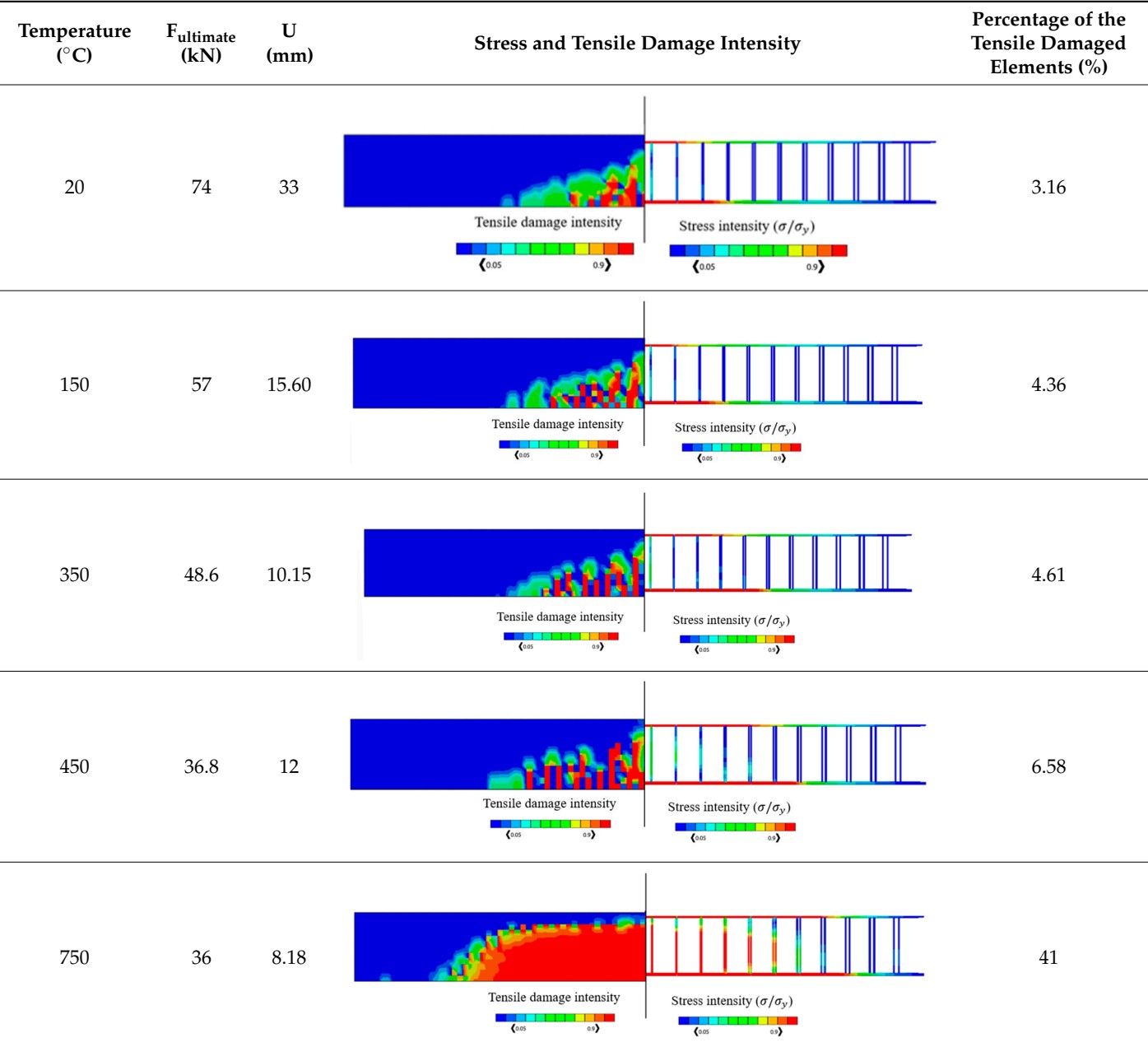

| Temperature (°C) | $F_{ultimate}$ (kN) | U (mm) | Stress and Tensile Damage Intensity | Percentage of the Tensile Damaged Elements (%) |
|---|---|---|---|---|
| 20 | 74 | 33 | | 3.16 |
| 150 | 57 | 15.60 | | 4.36 |
| 350 | 48.6 | 10.15 | | 4.61 |
| 450 | 36.8 | 12 | | 6.58 |
| 750 | 36 | 8.18 | | 41 |

**Table 5.** Probabilistic parameters of the concrete material.

| Parameter | Unit | Distribution | Mean Values | Coefficient of Variation | Source |
|---|---|---|---|---|---|
| Compressive strength ($f'c$) | MPa | Normal | Table 3 | 10% | [56] |
| Young's modulus ($E_0$) | GPa | Normal | Table 3 | 8% | [56] |

Tables 6–10 represent the obtained results of the probabilistic analysis of different considered temperatures, where each table shows a comparison between the resulting load, displacement, stress and tensile damage intensity, and the percentage of the damaged elements within the model according to the values of the reliability index ($\beta$) for each temperature case. It can be noted for each temperature case that as ($\beta$) increases, the

corresponding load and displacement values decrease. For instance, by considering the case of temperature = 20 °C, the displacement value is decreased by 17.92% from 13.00 mm when $\beta$ = 3.05 to 10.67 mm when $\beta$ = 3.40. Furthermore, it is also noted that the red zones of the model for each temperature case decrease as ($\beta$) increases, which means less plastic behavior in the beam. To be more specific, by considering temperature = 20 °C, the percentage of the tensile damaged elements was decreased from 3.47% when $\beta$ = 3.05 to 2.53% when $\beta$ = 3.40. Furthermore, in the case of temperature = 750 °C, the damaged elements percentage was decreased from 28.69% when $\beta$ = 3.05 to 26.53% when $\beta$ = 3.40.

**Table 6.** Probabilistic results—temperature = 20°C.

| $\beta$ | F (kN) | U (mm) | Stress and Tensile Damage Intensity | Percentage of the Tensile Damaged Elements (%) |
|---|---|---|---|---|
| 3.40 | 67.4 | 10.67 | 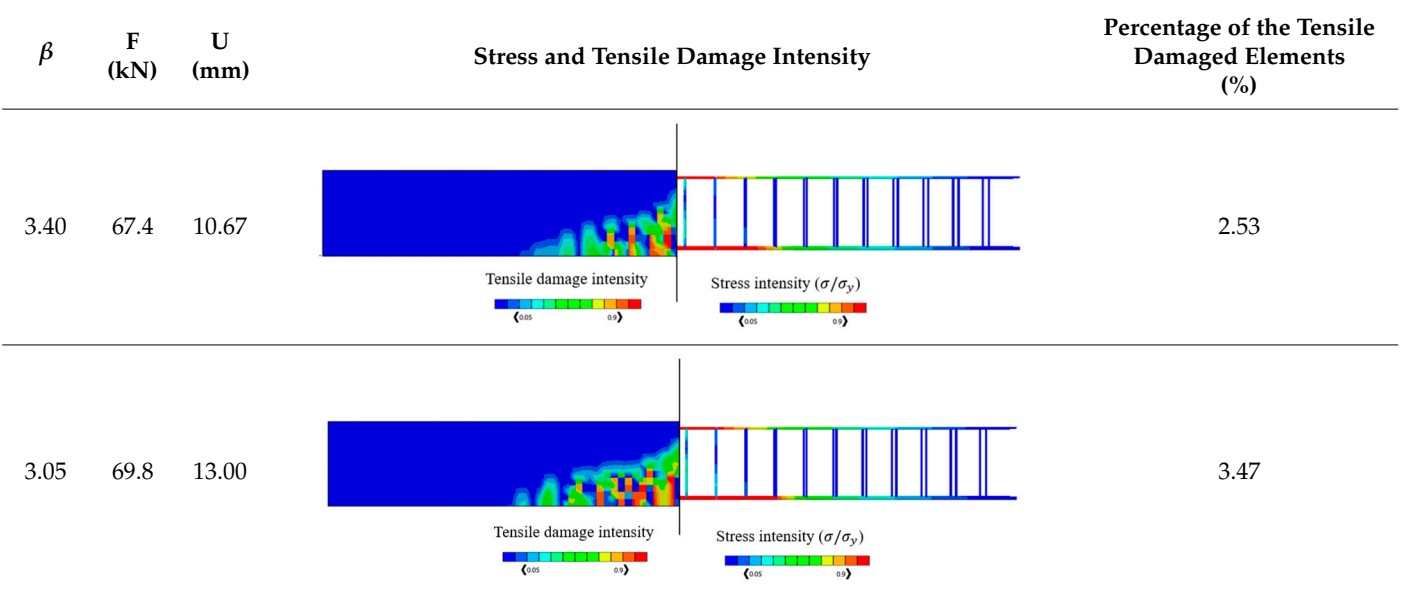 | 2.53 |
| 3.05 | 69.8 | 13.00 | | 3.47 |

**Table 7.** Probabilistic results—temperature = 150°C.

| $\beta$ | F (kN) | U (mm) | Stress and Tensile Damage Intensity | Percentage of the Tensile Damaged Elements (%) |
|---|---|---|---|---|
| 3.40 | 52.6 | 5.09 | 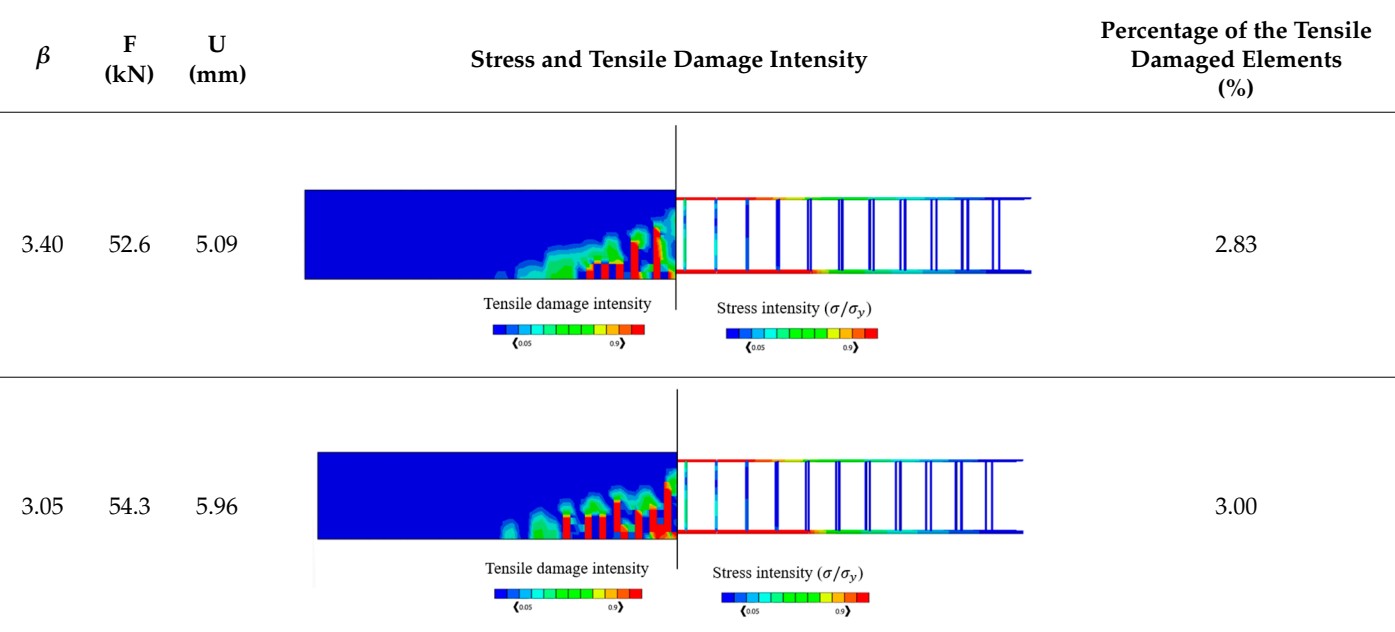 | 2.83 |
| 3.05 | 54.3 | 5.96 | | 3.00 |

**Table 8.** Probabilistic results—temperature = 350°C.

| $\beta$ | F (kN) | U (mm) | Stress and Tensile Damage Intensity | Percentage of the Tensile Damaged Elements (%) |
|---|---|---|---|---|
| 3.40 | 44.7 | 5.11 | | 3.28 |
| 3.05 | 46.5 | 5.56 | | 3.44 |

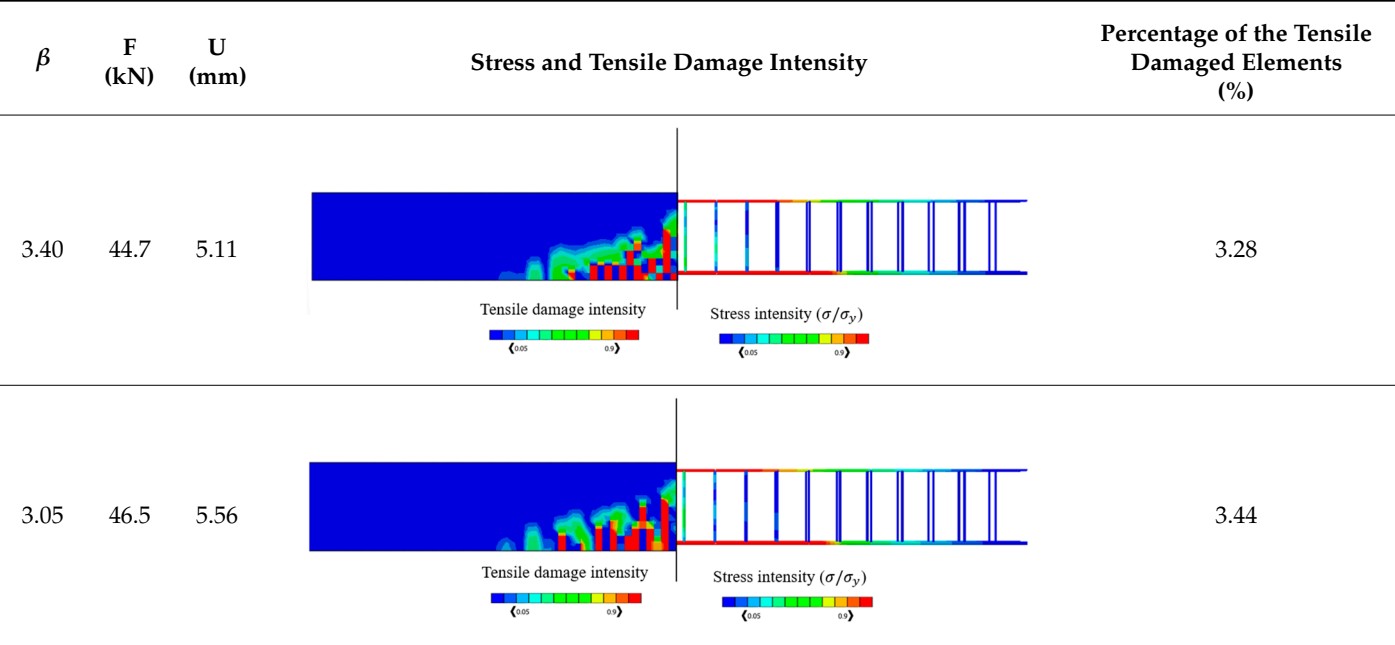

**Table 9.** Probabilistic results—temperature = 450°C.

| $\beta$ | F (kN) | U (mm) | Stress and Tensile Damage Intensity | Percentage of the Tensile Damaged Elements (%) |
|---|---|---|---|---|
| 3.40 | 32.1 | 4.94 | | 5.64 |
| 3.05 | 33.8 | 6.02 | | 6.08 |

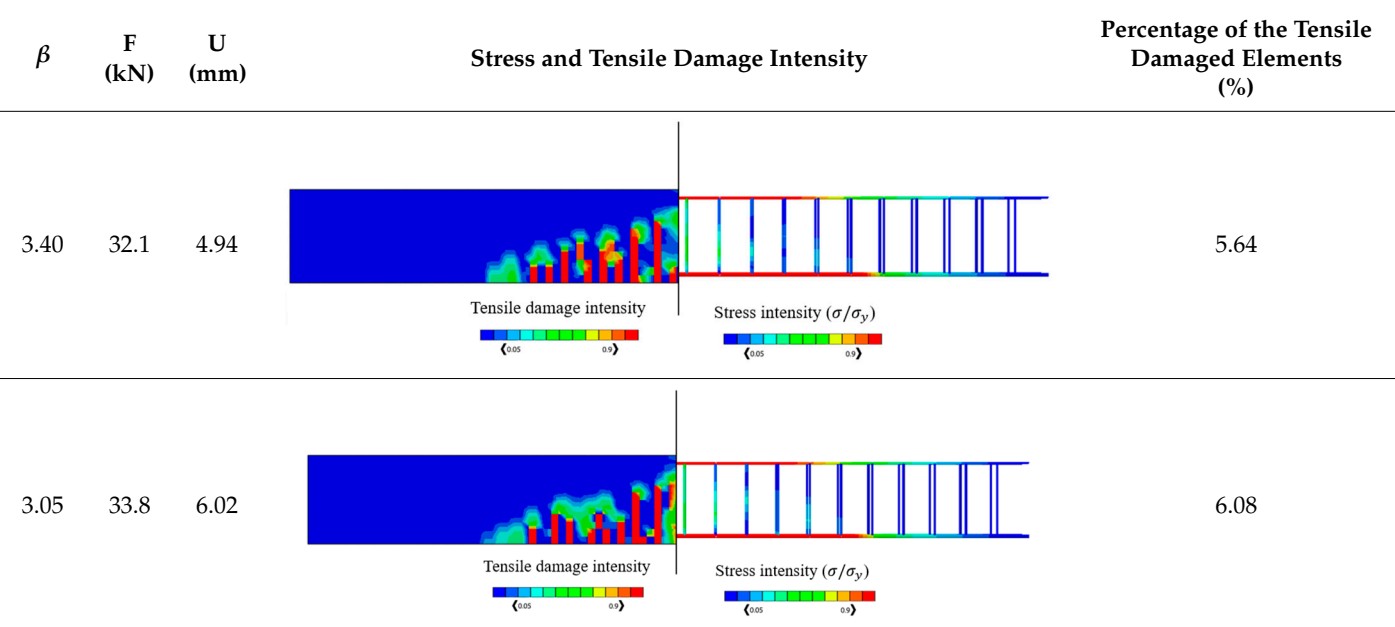

According to the results of reliability-based design, we can say that in the case of high temperatures, the percentage of damaged zones within the model is higher than those which were observed in lower temperature cases. Furthermore, the considerations of mechanical properties of concrete and steel materials as random variables successfully proved that the results are changed in the probabilistic design compared to the deterministic designs according to the resulting displacement, load, and the tensile damage and the stress intensities. Therefore, we can say that ($\beta$) efficiently worked as a limit to control the nonlinear plastic state of the models.

**Table 10.** Probabilistic results—temperature = 750°C.

| β | F (kN) | U (mm) | Stress and Tensile Damage Intensity | Percentage of the Tensile Damaged Elements (%) |
|---|---|---|---|---|
| 3.40 | 31 | 6.57 | 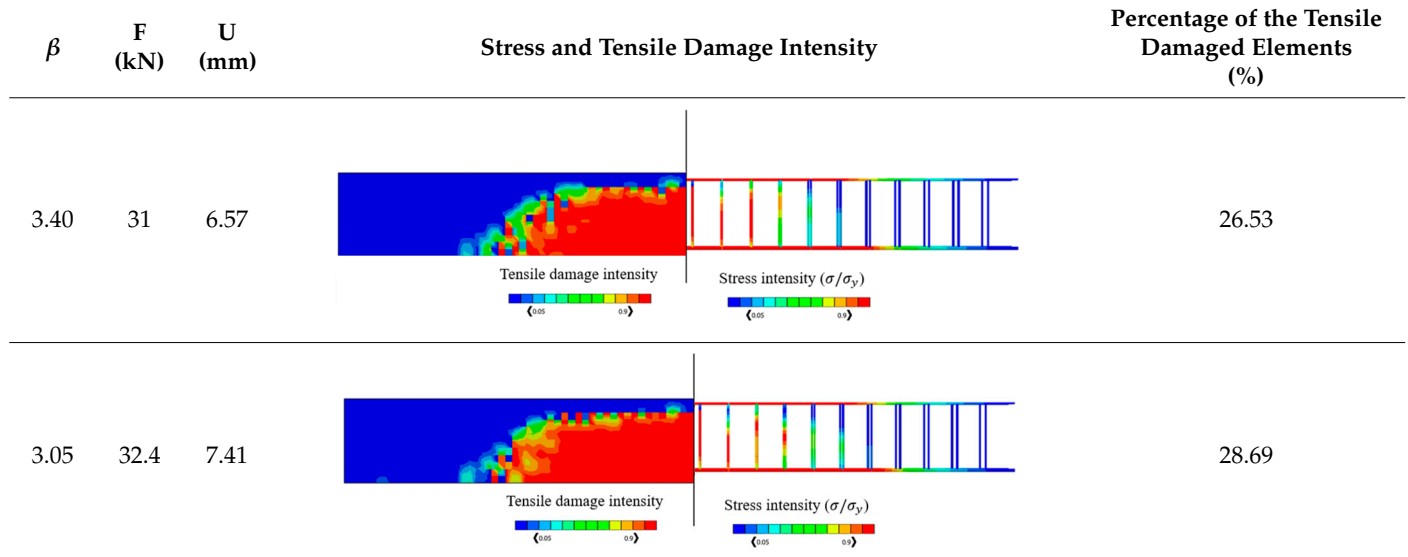 | 26.53 |
| 3.05 | 32.4 | 7.41 | | 28.69 |

## 6. Conclusions

In this paper, reliability nonlinear finite element models were considered for analyzing reinforced concrete beams at elevated temperatures. A concrete damage plasticity constitutive model was adopted to calibrate the numerical model according to generated data from laboratory tests. Moreover, the procedure of introducing the reliability analysis of the nonlinear mathematical problems was proposed by a nonlinear written code considering different reliability index value for each temperature case. This took into consideration that the concrete and steel materials' properties were considered random variables with a mean value and standard deviation.

Accordingly, the main points which conclude the proposed work are as follows:

1. In cases of deterministic design, when temperature is increased, the damage pattern, and stress intensity distributions are extended away from the middle area of the model.
2. For all models with different temperature cases, it was shown that by considering $\beta$, the corresponding loads and displacements were changed from the resulting values in deterministic designs due to considering concrete properties as random variables.
3. The intensity of the tensile damage pattern and intensity of stresses in cases of reliability-based design are less than was observed in cases of deterministic models for each temperature case.
4. The results showed that as $\beta$ increases, the corresponding load and displacement values decrease for each temperature case in the case of probabilistic analysis.
5. The pattern of tensile damage and the stress intensities become less intensive as $\beta$ increases for each temperature case in the case of probabilistic approach. Therefore, $\beta$ can work as a controlling bound for producing a safe plastic design.

Based on the findings of this study, the following recommendations for practice and prospective work are suggested:

1. Design codes and guidelines for reinforced concrete structures should incorporate probabilistic approaches to account for the uncertainties in material properties and loading conditions. This can help to ensure more reliable and safer designs.
2. The proposed approach can be extended to other types of reinforced concrete structures, such as columns and slabs, to investigate their behavior under elevated temperatures.
3. Future research can focus on investigating the effect of other parameters on the reliability of reinforced concrete structures at high temperatures, such as the effect of different types of reinforcements, different loading conditions, and the effect of cooling methods.

4. The developed approach can be combined with other methods, such as fire resistance tests, to validate and improve the accuracy of the numerical models.

By implementing these recommendations, it is possible to advance the design and analysis of reinforced concrete structures under elevated temperatures and enhance their safety and reliability.

**Author Contributions:** Methodology, J.S. and M.M.R.; conceptualization, J.L. and M.M.R.; investigation, M.M.R.; writing—original draft preparation, M.H. and M.M.R.; software, M.H.; writing—review and editing, J.S.; formal analysis, M.H.; validation, J.S.; supervision, J.L. All authors have read and agreed to the published version of the manuscript.

**Funding:** This research received no external funding.

**Data Availability Statement:** The datasets created and analyzed during the present work are provided in the main publication; more information is available from the authors.

**Conflicts of Interest:** The authors state that they have no known conflicting financial interest or personal relationship that might seem to have influenced the research presented in this study.

## Nomenclature

| Variable | Full Variable Description |
|---|---|
| $\overline{\sigma}_c$ | The effective compression internal force |
| $\overline{\sigma}_c$ | The effective uniaxial compressive stress |
| $\overline{\sigma}_{ij}$ | Effective internal force |
| $\hat{\sigma}_{ij}$ | Internal force |
| $\overline{\sigma}_t$ | The effective tension internal force |
| $\overline{\sigma}_t$ | The effective uniaxial tensile stress |
| $D_0^{el}$ | Elastic stiffness of the material |
| $D_f$ | Failure domain |
| $D_{ijkl}^{el}$ | Degraded elastic stiffness |
| $E_0$ | The initial Young's modulus |
| $P_f$ | Probability of failure |
| $X^{(z)}$ | Independent random vectors |
| $d_c$ | Variable of compression damage |
| $d_t$ | Variable of tension damage |
| $f_X(x)$ | Probability density function |
| $\varepsilon_c$ | Compressive strain |
| $\varepsilon_c^{pl,h}$ | The equivalent compression plastic strains |
| $\varepsilon_{ij}$ | Strain tensor |
| $\varepsilon_{ij}^{el}$ | Elastic part of strain tensor |
| $\varepsilon_{ij}^{pl}$ | Plastic part of strain tensor |
| $\varepsilon_t$ | Tensile strain |
| $\varepsilon_t^{pl,h}$ | The equivalent tension plastic strain |
| $d$ | Stiffness degradation |
| $f_{b0}/f_{c0}$ | The ratio of initial equi-biaxial compressive yield stress to initial uniaxial compressive yield stress |
| $K$ | Softening parameter |
| $z$ | Number of sample points |
| $\mathbb{E}$ | Mean value |
| $\mathbb{V}ar$ | Variance |
| $\beta$ | Reliability index |

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
