# Peer review of "Reliability Assessment of Reinforced Concrete Beams under Elevated Temperatures: A Probabilistic Approach Using Finite Element and Physical Models"

_sustainability, doi:10.3390/su15076077_

Round 1

Reviewer 1 Report

In this interesting paper, the authors presented reliability and FE models were considered for of RC concrete beams at elevated temperatures. In their study, a CDP constitutive model was adopted to calibrate the numerical model according to generated experimental data. Additionally, the procedure of introducing the reliability analysis of the nonlinear mathematical problems were proposed by a nonlinear written code considering different reliability index value for each temperature case. While their technical writing is acceptable for their first submission, they still need to proofread it carefully for the final submission. Overall, the work is interesting and the methodology is plausible. A number of technical and organizational concerns must also be addressed prior to reconsideration. The following is a summary of my assessment of the article.

-          The paper's title does not accurately reflect its scope because FE and Physical models are not included. Please consider modifying it.  

-          Expand on the literature review by including some information from:

1)      Eamon, C.D. and Jensen, E., 2013. Reliability analysis of RC beams exposed to fire. Journal of Structural Engineering139(2), pp.212-220.

2)      Li, Z., Xiao, J. and Xie, Q., 2021. Reliability analysis of the residual moment capacity of high‐strength concrete beams after elevated temperatures. Structural Concrete22(3), pp.1586-1599.

3)      Song, Y., Fu, C., Liang, S., Yin, A. and Dang, L., 2019. Fire resistance investigation of simple supported RC beams with varying reinforcement configurations. Advances in Civil Engineering2019.

4)      Cai, B., Li, B. and Fu, F., 2020. Finite element analysis and calculation method of residual flexural capacity of post-fire RC beams. International Journal of Concrete Structures and Materials14, pp.1-17.

5)      Agrawal, A. and Kodur, V., 2019. Residual response of fire‐damaged high‐strength concrete beams. Fire and Materials43(3), pp.310-322.

6)      Yuye, X., Bo, W., Ronghui, W., Ming, J.I.A.N.G. and Yi, L.U.O., 2013. Experimental study on residual performance of reinforced concrete beams after fire. Journal of Building Structures34(8), p.20.

7)      Esfahani, M., Hoseinzade, M., Shakiba, M., Arbab, F., Yekrangnia, M. and Pachideh, G., 2021. Experimental investigation of residual flexural capacity of damaged reinforced concrete beams exposed to elevated temperatures. Engineering Structures240, p.112388.

8)      Van Cao, V., 2022. Reliability-Based Moment Capacity Assessment of Reinforced Concrete Beams in Fire. International Journal of Civil Engineering20(11), pp.1291-1308.

9)      Kodur, V.K.R., Dwaikat, M.B. and Fike, R.S., 2010. An approach for evaluating the residual strength of fire-exposed RC beams. Magazine of Concrete Research62(7), pp.479-488.

10)  Dao, D.V., Adeli, H., Ly, H.B., Le, L.M., Le, V.M., Le, T.T. and Pham, B.T., 2020. A sensitivity and robustness analysis of GPR and ANN for high-performance concrete compressive strength prediction using a Monte Carlo simulation. Sustainability12(3), p.830.

11)  Abid, S.R., Abbass, A.A., Murali, G., Al-Sarray, M.L., Nader, I.A. and Ali, S.H., 2022. Repeated impact response of normal-and high-strength concrete subjected to temperatures up to 600 C. Materials15(15), p.5283.

12)  Cho, H.C., Han, S.J., Heo, I., Kang, H., Kang, W.H. and Kim, K.S., 2020. Heating temperature prediction of concrete structure damaged by fire using a bayesian approach. Sustainability12(10), p.4225.

-          The problem in this study is not fully understood, as there is no discussion environment for the references cited in the organization of the introduction.

-          In the reliability analysis, it is not clear how  (Equation (2)) were evaluated. Usually this state limit function needs a formula for   and evaluation of a professional factor that validated against robust data. The authors need to expand about this unclear information.

-          In the Experimental tests section, please provide the details of the materials, method testing (compressive and tensile behavior of concrete and reinforcing steel.). Also, Please explain more details about design procedure of the beam specimens.

-          Details of the software used for the numerical analysis should be given.

-          In table (2), Which was the criterion used to define the dilation angle for the analyzed beams?

-           

-          Please provide tensile model adopted in the numerical simulation. Table 2 provides the CDP model for the compressive behavior of concrete.

-          Enhance the quality of Figure 3 (sectional dimensions are not clear) Figure 7 and Figure 8.

-          In Figure 3, show the load-deflections curves for all the test specimens (not only the average). This will give an idea bout he repeatability of the test results.

-          The authors should justify the design of the test specimens, and justify the number of experiments. What are the limitations of the study? If you change the mix design, would your conclusion still hold true?

-          Recommendations for practice and prospective work is missing.

-          The authors should include a list of notations at the end of the paper, as many equation are present (27).

For the reasons noted above, I believe the manuscript needs to undergo substantial review before being considered for publication in this prestigious journal.

Reviewer 2 Report

This paper proposes a computational model to analyze the reliability of reinforced concrete beams at high temperatures. The model considers uncertainties in the mechanical properties of concrete and steel materials and uses the reliability index as a constraint. The results show the critical role of uncertainties in the modelling process.

This study shows some merit; however, it needs improvement regarding the numerical model (FE) and the presentation for better readability.  The authors claimed that they present a novel computational model, which is somehow an exaggeration. In fact, the novelty of the work is not so evident. 

The study is based on the reliability analysis of the concrete at hot temperature; unfortunately, only the experimental tests carried out by the authors are at room temperature, and the results at hot temperature are obtained from the literature (Eurocode).

The reviewer reveals some weaknesses of the manuscript and recommends to carefully be addressed by the authors:

1) The main concern is the FE model, for which it is not clearly shown in the manuscript how the temperature was taken into account in the simulation? What kind of Finite element should be chosen to consider the the temperature. 

2) The simulation at hot temperature requires the definition of the thermal properties of the deformed media (concrete and steel). No indication in the manuscript given on the thermal capacity of the materials, etc...

3) To better compare the results obtained using the deterministic and the reliability analysis, Table 5 should be combined with Table 4 and Table 6 to obtained similar representation as Table 8.

4) The figure captions should be improved and adequately detailed for better clarity.

5) In section 5, the first subsection should be 5.1 Finite element model and the first paragraph on page 8, subsection 5.2 Model validation should be used. 

I recommend that the paper be revised to address the weaknesses mentioned above before it is considered for publication.

Reviewer 3 Report

This paper presents a novel reliability approach to r/c beams considering elevated temperatures.  The whole text is very well-written and scientifically based.

Please check the size of fonts in pages 11-12.  Also, please add which is  the used FEA software.  

Round 2

Reviewer 2 Report

In the revised version of the paper, the authors have responded in satisfied manner to the comments of the reviewer. The the reviewer suggests the publication of manuscript in Sustainability Journal.

Author Response

The authors express their sincere thanks and appreciation to the editor and reviewer for taking the time to review this manuscript.

With Best Regards.